# SCHEDULING DATA IMPROVES FINE-TUNING DATA EFFICIENCY

## ABSTRACT

To train a language model for a target domain with a limited amount of data (e.g. math), the current paradigm is to pre-train on a vast amount of generic web text and then fine-tune on the target data. Since standard fine-tuning uses a data schedule of keeping all generic data before all target data, we ask how much we can improve performance on the target domain via adding generic data to the end of training or target data to the start. In a controlled pre-training environment, we first show that simply replaying generic data while fine-tuning, though typically used to reduce catastrophic forgetting of the generic domain, can surprisingly improve performance on the target domain. We then merge the two stages of pre-training and fine-tuning into a single learning rate schedule to establish a mid-training baseline that better leverages the target data. Under this merged learning rate schedule, we search over two stage data schedules that additionally move target data earlier in training. After composing our three interventions, we estimate that standard fine-tuning would need up to $15.86\times$ more data to match the target performance of our best data schedule. We test our findings at scale by showing how replay improves performance for larger models on downstream tasks, improving agentic web navigation success by $4.5\%$ and Basque question-answering accuracy by $2\%$.

## 1 INTRODUCTION

To train a language model for a target domain with a limited amount of data (e.g. math, instruction following), current practice often pre-trains a language model on a vast amount of generic web text before fine-tuning on the target data (Hernandez et al., 2021; Ouyang et al., 2022). Since standard fine-tuning uses a data schedule of keeping all generic data before all target data, we ask how much we can improve performance on the target domain via mixing generic data at the end of training or mixing target data at the start of training. In this work, we show evidence that both such interventions can significantly improve performance on the target domain, better leveraging the limited data.

First, we explore whether introducing generic data at the end of training can actually improve performance on the target domain (Section 3). We start our investigation in a controlled pre-training environment with two pools of data: generic web pre-training data (i.e. C4) and target data from a domain of interest (i.e. FineMath, StarCoder, and Flan instruction following). In this setting, we tune a competitive standard fine-tuning baseline pre-training on solely generic data and fine-tuning on solely target data according to common practice (e.g. separate learning rate schedules and optimizer states). In this setting, generic data is sometimes mixed at the end of training to prevent catastrophic forgetting of the generic domain. However, we surprisingly find that replay can improve performance on the target domain even though the generic data pushes the training distribution at the end further from the target domain, improving data efficiency by up to $1.49\times$ for FineMath.

Next, we determine whether we can improve target performance by moving target data to the start of training (Section 4). Since we are now allowed to change pre-training in service of the target domain, we use a single learning rate schedule with Warmup-Stable-Decay (WSD) (Hu et al., 2024) following practice in mid-training (Grattafiori et al., 2024; OLMo et al., 2025; Li et al., 2025). WSD does not reset the optimizer state during training and only anneals learning rate once sharply at the end of training. The mid-training baseline is already $6.37\times$ more data efficient than standard fine-tuning for FineMath, suggesting that model developers should release checkpoints prior to annealing to improve adaptation with target data. After tuning mid-training, we consider two stage data schedules which,

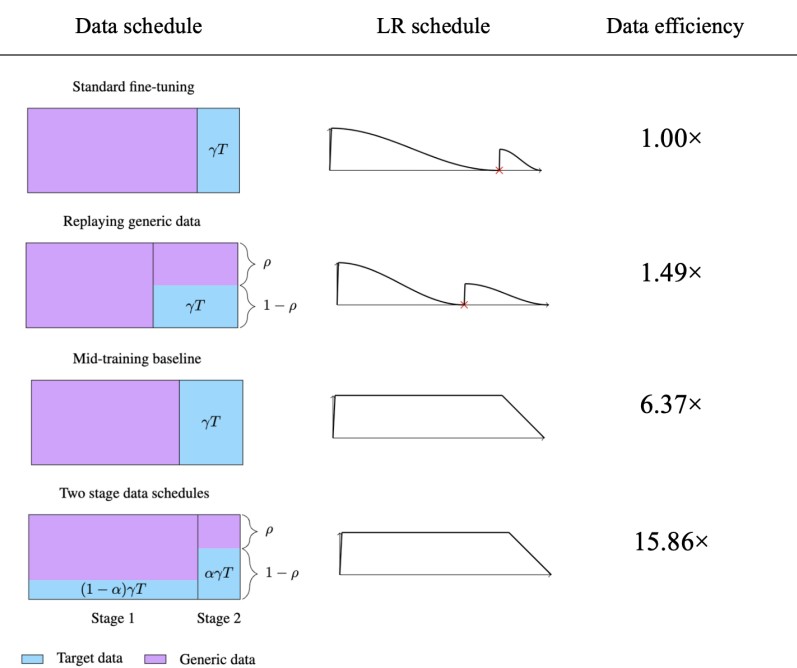

Figure 1: **Improving data efficiency by changing the fine-tuning data schedule.** Since pre-training on generic data and fine-tuning on target data is the de facto data strategy to leverage target data, we study whether we can improve upon its data schedule. We first find that by replaying generic data at the end of training can surprisingly improve performance on the less relevant target domain. We then turn to merging the optimization of both training stages by using a single WSD learning rate schedule. Finally, we additionally introduce target data during stage 1. Combining these interventions results in a $15.86\times$ data efficiency improvement over standard fine-tuning for the target data.

in addition to replaying generic data, use target data earlier in training. Our best data schedule is up to $15.86\times$ more data efficient than standard fine-tuning. Interestingly, we find that increasing the replay fraction is generally most important when there is no target data in the first stage.

We test whether our findings hold beyond our controlled experimental setting by utilizing replay data when fine-tuning larger models (i.e. Llama 3 8B Base and Instruct) on real target tasks. Since our two stage data schedule findings suggests that replay helps the most when target domains that are underrepresented in pre-training, we test whether replay helps for such domains. We find that replay improves performance on agent benchmarks with limited trajectories (increasing web navigation success rate by $4.5\%$) and improves low-resource language learning from limited documents (increasing Basque question-answering accuracy of Llama 3 8B Instruct by $2\%$).

## 2 CONTROLLED PRE-TRAINING SETUP

Our goal is to search for data schedules that outperform standard fine-tuning. However, pre-training at the scale of frontier models is prohibitively expensive. Therefore, we study this question by establishing a controlled pre-training setup with stylized simplifications that enable science at a smaller scale. We stress test our conclusions at scale in Section 5.

### 2.1 DATA AND TRAINING

To model a natural fine-tuning setting, we build a pool of **generic data** representing standard internet web text for pre-training and **target data** representing a domain of interest. In our experiments, we use C4 for our generic domain and FineMath (math), StarCoder (coding), and Flan (instruction following) for our target domains. In addition to representing unique downstream tasks, these domains abstractly

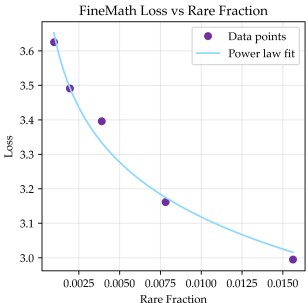

Figure 2: **Data scaling law for reference algorithm.** We run a reference training strategy with different target data budgets. To estimate how effectively an algorithm is using the data, we invert the reference strategy's scaling law to recover "effective data" for this loss and compare the data efficiency improvement between two strategies.

reflect different levels of similarity with the generic data: StarCoder is furthest from the generic data since C4 is filtered for code whereas Flan is closest since it contains the most natural language.

Since web data is abundant relative to target data, we do not constrain the amount of generic data and instead constrain the total number of training steps to total 4 billion tokens for compute-matched comparisons. We model a data constraint on the amount of target data, typically taken to be 4 million tokens. We follow a strong existing recipe for pre-training a 150 million parameter Llama-style language model (Grattafiori et al., 2024) with AdamW, with full training details in Appendix D.

## 2.2 EVALUATION

We are interested only in performance on the target domain, which we measure via loss on a held-out validation set from the target distribution. We choose validation loss since it scales much more smoothly than accuracy metrics for models at our scale and is known to correlate with downstream performance (Thrush et al., 2025; Gadre et al., 2024; Chen et al., 2025c).

To compare training strategies, we define "data efficiency" to capture how effectively a training strategy is using the samples from the target domain. We formalize a training strategy S as accepting $D$ target tokens and producing a model with loss $\mathcal{L}(\mathrm{S}(D))$. To contextualize the importance of a loss improvement, we first measure the loss of a fixed reference strategy $\mathrm{S}_{\mathrm{ref}}$ for different target data budgets $D$. We then fit a scaling law that predicts the loss of the reference algorithm for $D$ tokens as $\hat{\mathcal{L}}_{\mathrm{ref}}(D)$, as visualized in Figure 2. To evaluate a training strategy S, we can estimate the effective target data the reference strategy would need to match the loss of S with $D$ tokens as $\hat{\mathcal{L}}_{\mathrm{ref}}^{-1}(\mathcal{L}(\mathrm{S}(D)))$. To remove this quantity's dependence on the data efficiency of the reference strategy, we report data efficiency as a relative improvement of $\mathrm{S}_2$ over $\mathrm{S}_1$, or $\frac{\hat{\mathcal{L}}_{\mathrm{ref}}^{-1}(\mathcal{L}(\mathrm{S}_2(D)))}{\hat{\mathcal{L}}_{\mathrm{ref}}^{-1}(\mathcal{L}(\mathrm{S}_1(D)))}$. Therefore, a data efficiency improvement of $k\times$ can be interpreted as "$\mathrm{S}_1$ would require $k$ times more target data to match the loss of $\mathrm{S}_2$ at $D$ tokens". We give more details on how we fit the scaling laws in Appendix H.

## 3 IMPROVING TARGET DATA EFFICIENCY OF FINE-TUNING

In this section, we study how much we can improve data efficiency by mixing generic data at the end of training. We consider data schedules with two stages: Stage 1 constitutes pre-training on only generic data and Stage 2 constitutes training on target data (potentially mixed with generic data). After establishing a competitive standard fine-tuning baseline (Section 3.1), we make the surprising observation that mixing generic data in Stage 2 improves target val loss (Section 3.2).

### 3.1 FINE-TUNING BASELINE

We first establish a competitive baseline to reflect standard fine-tuning. To define our data schedules, suppose we train for a total of $T$ steps, with $\gamma$ fraction of the steps on the target data. Standard fine-tuning corresponds to training on generic data with a cosine learning rate schedule for $(1 - \gamma)T$ steps, followed by training on the target data for $\gamma T$ steps with a separate cosine learning rate schedule. To match common practice for fine-tuning models, we reset the optimizer state (i.e. for AdamW, the estimate of the first/second moments of the gradients) in between the stages. We tune the two main choices for our baseline: learning rate and the target data epochs (exact procedure in Appendix G.1).

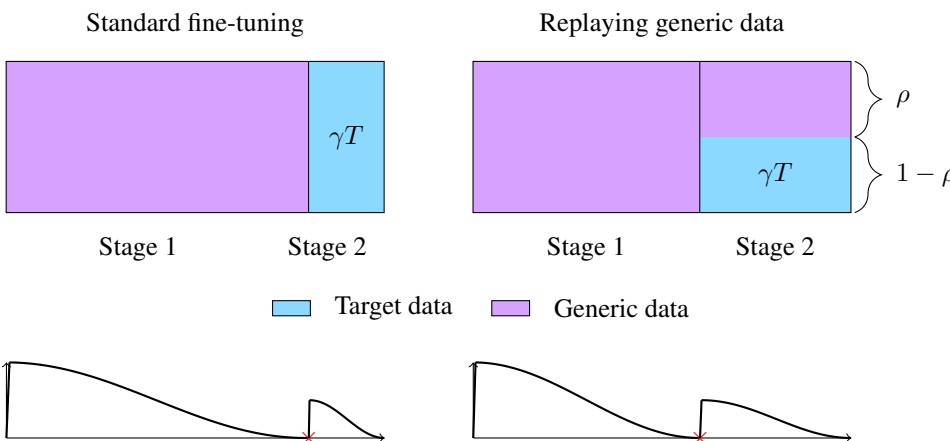

Figure 3: **Controlled fine-tuning visualization.** We systematically explore the benefit of replaying generic data while fine-tuning on the target data. On the right, we show standard fine-tuning for $T$ steps where $\gamma$ fraction of the steps are on target data. On the left, we show fine-tuning with replay fraction $\rho$ (where we shorten pre-training to keep the total number of steps fixed). We use (independently tuned) cosine learning rate schedules for each stage, with an optimizer state reset between the stages to simulate standard practice for fine-tuning open-weight models.

We find that if we try to repeat the data past a certain epoch count, the validation loss increases, akin to classical overfitting. This is not captured by the functional form of prior data-constrained scaling laws, discussed in Appendix L.1. This setup defines $1\times$ target data efficiency per domain.

## 3.2 REPLAY IMPROVES DATA EFFICIENCY

We introduce a simple strategy that improves loss on the target task: mix generic data while training on the target data. Specifically, we introduce a replay fraction $\rho$ for what fraction of training steps during Stage 2 will be on generic data. When we increase this replay fraction, we decrease the number of steps taken during Stage 1 steps to conserve the total number of training steps (Figure 3, right). In Figure 4, we show how the final model's loss depends on the replay fraction. We find that for each domain, a non-zero replay fraction minimizes the loss (indicated by the starred points), achieving a data efficiency of $1.87\times$ for Flan, $1.49\times$ for FineMath, and $1.09\times$ for StarCoder. We observe that code, which C4 explicitly filters out, can tolerate less replay data than the higher overlap domains of math and instruction following, though the loss improvements are similar across domains.

Though replay is a common method in continual learning, it is almost always used to prevent catastrophic forgetting of old tasks (Rolnick et al., 2019; Parisi et al., 2019). Interestingly, we find that replay improves performance on the new in-distribution training task, departing from the standard intuition. We provide a more detailed discussion in Section 6.

## 4 IMPROVING TARGET DATA EFFICIENCY OF PRE-TRAINING

In the previous section, we limited ourself to using replay during Stage 2. In this section, we aim to understand how much additional data efficiency we get by introducing target data during Stage 1, controlling the data mixture for both stages of training. We first unify the optimization process of pre-training and fine-tuning into a single learning rate schedule cycle with no optimizer state reset (Section 4.1). We then search over possible data schedules by choosing a replay fraction for Stage 2 as well as how much of the target data is seen in Stage 2 vs Stage 1, defined in Section 4.2. This extra flexibility allows us to quantify the benefit of introducing target data earlier in training during Stage 1 instead of reserving it all for the end during Stage 2. In Section 4.3, we show that mixing in target data during Stage 1 can strongly alleviate the need for replay. However, for one of our three domains, correctly tuned replay removes the need to introduce target data early during pre-training.

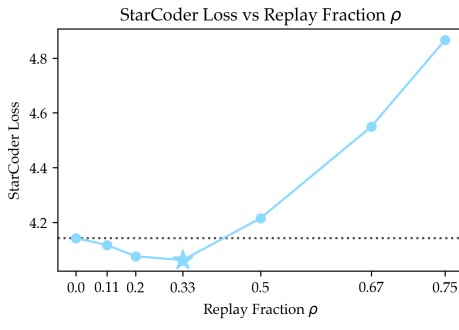

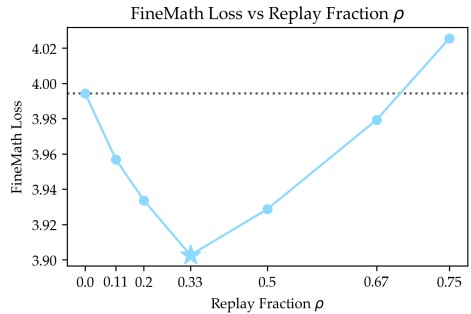

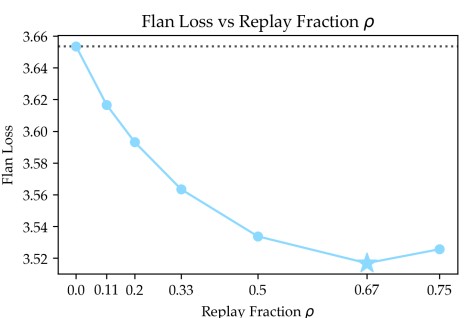

Figure 4: **Replay improves loss on target data.** We show that across our target domains, the correct amount of replay (starred points) beats the no replay baseline (dotted line). Though data distributions closer to pre-training (Flan) can tolerate more replay compared to further domains (StarCoder), the loss improvement is relatively constant across domains.

## 4.1 MID-TRAINING BASELINE

Now that we are allowed to change pre-training, we establish a mid-training baseline that outperforms standard fine-tuning. Similar to before, we tune the learning rate and epoch count (Appendix E.1.1). However, we find that learning rate schedule is critical for target data efficiency. Default practice (i.e. cosine, linear) is to slowly anneal to zero over the course of training. Recent work in mid-training instead suggests using a warmup-stable-decay (WSD) learning rate schedule (Hu et al., 2024). This consists of a short linear warmup, a stable training phase, and a sharp linear decay for a variable fraction of training referred to as the cooldown period. Interestingly, during the learning rate decay, the loss decreases at a much faster rate than the rest of training. This can be exploited to get stronger performance on target data by placing it at the end of training (Grattafiori et al., 2024; OLMo et al., 2025). We explain and visualize these benefits in more detail in Appendix I. In Figure 5, we show that WSD offers a significant benefit over traditional schedules that decay the learning rate over all of training. For our setting, we find it best to fix a cooldown period of 10% of training for all domains, increasing data efficiency by $28.47\times$ relative to the worst cooldown duration. We share more details on the learning rate in Appendix E.1.2).

The new mid-training baseline strategy increases data efficiency relative to the standard fine-tuning baseline from Section 3.1 by $9.92\times$ for Starcoder, $6.37\times$ for FineMath, and $2.77\times$ for Flan. This is likely because the joint training doesn't reset optimizer state and rewarmup the learning rate. As such, we believe fine-tuning in practice would benefit from initializing at a pre-annealed pre-training checkpoint instead of the final checkpoint. We call on model developers to release the model and optimizer state before cooldown since this is more useful for downstream applications.

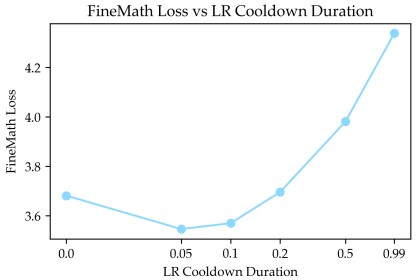

Figure 5: **Tuning learning rate cooldown.** We tune how long we should cool down the learning rate for WSD. The above plot shows the optimal cooldown period is between 0.05 and 0.1; we use 0.1 for consistency across domains and being fair to changing data schedules.

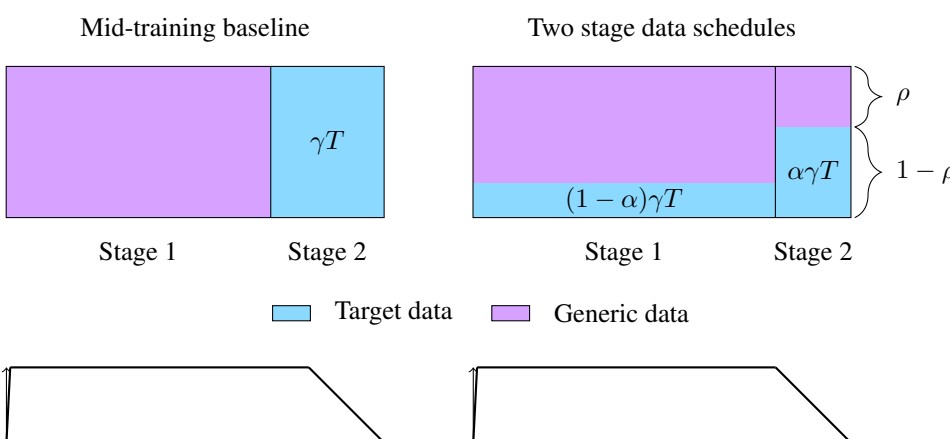

Figure 6: **Controlled mid-training visualization.** We explore the space of data schedules when training on $T$ tokens where a $\gamma$ fraction of the steps are on target data. A data schedule allocates an $\alpha$ fraction to Stage 2 where Stage 2 has a replay fraction $\rho$. Standard fine-tuning puts all target data at the end with no replay ($\alpha = 1$, $\rho = 0$). We use a WSD learning rate schedule across both stages.

## 4.2 DATA SCHEDULE SPACE

Given our mid-training baseline, we are interested in how much we can improve data efficiency by introducing target data at the start of training. Since it is too expensive to search over all possible permutations, we instead consider data schedules where we control the fraction of target data for each of two stages subject to the data constraint. This space now only has two degrees of freedom with multiple parameterizations. We decide to use the earlier notion of **replay fraction** $\rho$ (how much generic data is replayed during Stage 2) and introduce **target stage 2 allocation** $\alpha$ (what fraction of the total target data is allocated to Stage 2). We provide a more intuitive visualization in Figure 6. The data schedule for standard fine-tuning and the mid-training baseline have a simple interpretation: no replay data ($\rho = 0$) and allocating all target data to Stage 2 ($\alpha = 1$). Finding the optimal two stage data schedule now boils down to finding the best setting of $\rho$ and $\alpha$. We provide a detailed discussion of the parameterization and equivalences in Appendix A.

## 4.3 SEARCHING OVER TWO STAGE DATA SCHEDULES

We sweep over replay fraction $\rho$ and target stage 2 allocation $\alpha$ to find better data schedules. We are interested in three strategies: the mid-training baseline with the fine-tuning data schedule ($\rho = 0$, $\alpha = 1$), replaying generic data in Stage 2 ($\alpha = 1$), and the full space of modifications (all settings). We show the full results of sweeping over replay fraction and Stage 2 allocation in Figure 7. Pure fine-tuning is the top-right entry, fine-tuning with replay is the right column, and the full space of modifications is the entire plot. By only introducing generic replay, we find that we get data efficiency improvements over the mid-training baseline of $1.53\times$ for StarCoder, $1.85\times$ for FineMath, and $2.06\times$ for Flan. When searching over data schedules that also introduce target data in Stage 1, we find data efficiency improvements of $1.53\times$, $2.49\times$, and $4.80\times$ over the same baseline.

### 4.3.1 REPLAY IS MORE HELPFUL FOR DISSIMILAR STAGES

We find that replay is most helpful when the data in Stage 2 is most dissimilar from Stage 1. We first find that when the target data is unseen during Stage 1, replay is critical for improving loss (example for Starcoder in Figure 8, blue). On the other hand, when we keep 75% of the target data for pre-training, replay is no longer helpful (Figure 8, purple). Together, this indicates that replay is more helpful when the data in Stage 2 would otherwise be dissimilar to the data in Stage 1. This intuitively suggests that training benefits from a smooth transition when changing data distributions, whether it comes from adding generic data to Stage 2 or target data to Stage 1.

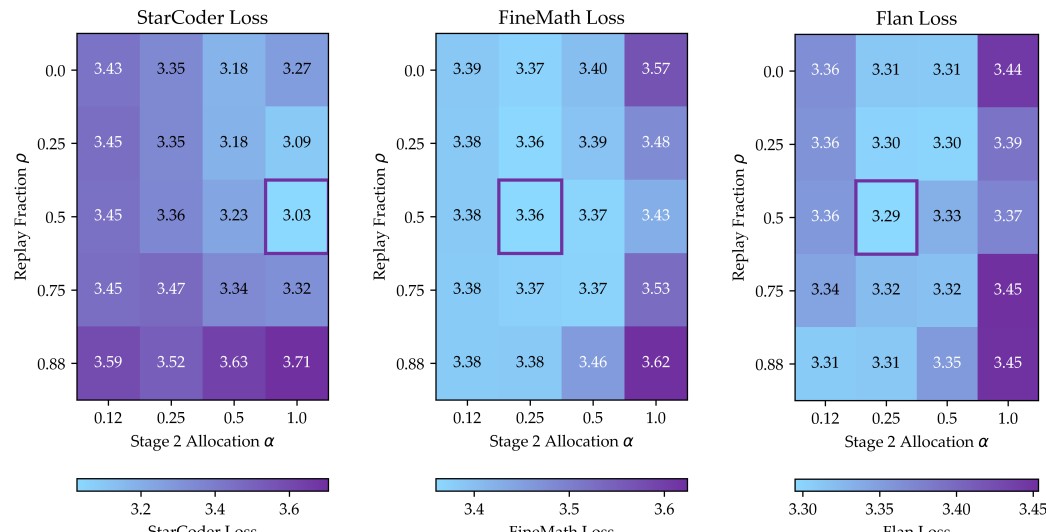

Figure 7: **Full data schedule sweep.** We sweep over data schedules, parameterized by their replay fraction and fraction of target data allocated to Stage 2. Standard fine-tuning (no replay and all target data in Stage 2, top right corner) achieves the worst loss for FineMath and Flan. This can be improved by adding replay data (right column). This can also be solved by adding some target data to Stage 1, in which case replay becomes less important.

### 4.3.2 DO WE NEED TO CHANGE PRE-TRAINING?

One natural question for applications is whether one needs to change pre-training (in this case, Stage 1) to maximally leverage task-relevant data. We find that for FineMath and Flan, we can not get the full benefits of the optimal data schedule by only changing Stage 2 (achieving $67.4\%$ of gains for FineMath and $46.0\%$ of the gains for Flan measured logarithmically with respect to the mid-training baseline). On the other hand, for Starcoder, the optimal data schedule only requires adding replay data to Stage 2. It is encouraging that we can get away with not using data early since it is prohibitive or impossible to change pre-training for many applications.

## 5 SCALING UP FINDINGS FOR POST-TRAINING

Since our conclusions so far have been developed in our controlled environment, we test whether our findings hold for larger language models under realistic training conditions. Following our analysis in Section 3, we expect replay to help performance on target domains. Following our analysis in Section 4, we expect this benefit to hold when the target data is relatively underrepresented during pre-training. Therefore, we test whether replay is helpful when fine-tuning strong language models (i.e. Llama 3)

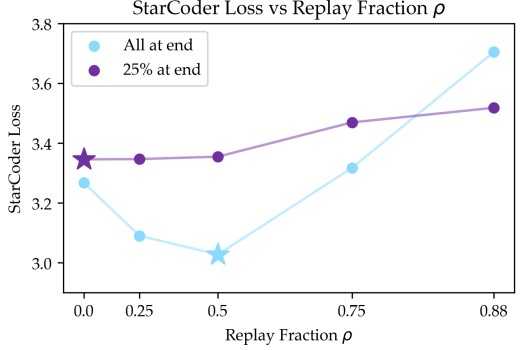

Figure 8: **Importance of replay fraction depends on rarity.** When all the target data is seen during fine-tuning, tuning the replay fraction becomes critical to improve loss (blue line). When we change pre-training to see some of the target data, tuning the replay fraction is not important and can sometimes hurt loss (purple line).

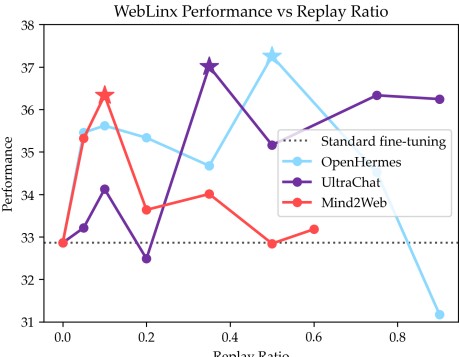

Figure 9: **Weblinx.** We fine-tune Llama 3.1-8B Instruct on Weblinx demonstrations. Without any replay data, we get 32.86% accuracy following the original hyper-parameters. We find that mixing generic instruction following data (OpenHermes, UltraChat) improves accuracy by up to 4.5%. This is even better than replaying demonstrations from an alternative web agent task (Mind2Web).

on target tasks such as web agent navigation (Section 5.1) and Basque language learning (Section 5.2). Though this sounds like a simple modification, this is rarely done while fine-tuning in practice as it is commonly believed that fine-tuning is the correct way to leverage target data.

**Setup.** We set up fine-tuning to best mimic standard practice. Since the pre-trained model is often fully annealed with no associated optimizer state, we follow the fine-tuning learning rate schedule used in Section 3. Moreover, since we usually do not have access to the generic data distribution, we have to pick an approximation of the data used in the previous training stage. We note that using a replay fraction of $\rho$ requires $\frac{1}{1-\rho}$ times as many training steps, which is generally permissible for fine-tuning since it is rarely compute-constrained.

## 5.1 WEB AGENTS

Recently, language models have been trained to perform agentic tasks, requiring capabilities such as web navigation can be learned from an expensive and limited number of human trajectories. We study this by following the supervised agent training strategy and evaluation established in Weblinx (Lù et al., 2024) when fine-tuning Llama 3.1 8B Instruct on a fixed number of target demonstrations. For the replay data, we use OpenHermes (Teknium, 2023) or UltraChat (Ding et al., 2023) instruction-following data to approximate the data distribution of the previous training stage.

We find that when training on web agents data under the hyperparameters from the original paper, there is a consistent advantage to replaying instruction following data under their offline scoring procedure. In Figure 9, we show that replaying instruction following data improves accuracy by up to 4.5%. We provide additional details/experiments in Appendix J.

## 5.2 BASQUE

Basque is a low-resource language constituting only 0.035% of Common Crawl (Etxaniz et al., 2024). However, thanks to a thriving NLP research community, there is a large amount of additional Basque data available through the Latxa corpus (Etxaniz et al., 2024). We are interested in how to continually pre-train Llama 3.1 8B with access to a limited number of Basque tokens (i.e. 200M). For replay data, we use the SlimPajama replication (Soboleva et al., 2023; Weber et al., 2024) as a proxy for the unreleased Llama pre-training data. For evaluation, we measure accuracy on a professional Basque translation (Baucells et al., 2025) of the commonsense reasoning benchmark COPA (Gordon et al., 2012; Ponti et al., 2020) supported on `lm-eval-harness` (Gao et al., 2024).

We find that when training on Basque data, there is a consistent advantage to replaying pre-training-like data. In Figure 10, we show that the model achieves higher accuracy on the Basque evaluation task. We provide more details and experiments in Appendix K.

## 6 RELATED WORK

We discuss additional related work in Appendix L.

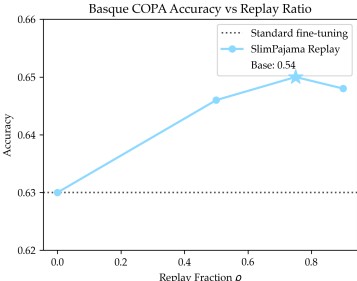

Figure 10: **Basque.** We fine-tune Llama 3.1-8B on 200M Basque tokens from the Latxa training corpus and measure accuracy on Basque COPA. We find that replaying generic pre-training data from SlimPajama improves accuracy by up to 2%.

**Mid-training.** Many recent language models augment pre-training with a mid-training phase that anneals the learning rate while training on high-quality data OLMo et al. (2025); Grattafiori et al. (2024); Li et al. (2025); Nvidia et al. (2024). There has been some initial work on characterizing the benefit of putting target data at the end of training (Aryabumi et al., 2024; Blakeney et al., 2024) or annealing the learning rate (Hu et al., 2024). In addition to prior knowledge, we conduct experiments on changing the pre-training in conjunction with mid-training. Moreover, we show new experiments at maximal repetition count, as well as for various ablation factors such as model size.

**Optimizing data mixtures.** Prior work has proposed algorithms to optimize the data mixture (Chen et al., 2023; Xie et al., 2023; Jiang et al., 2024a; Fan et al., 2024). Most such algorithms fall under an online optimization framework (Chen et al., 2025b), where the algorithm estimates which components it should upweight. However, such online algorithms are insufficiently myopic and miss that better data should be at the end. Instead, such algorithms greedily upweight the best data at the start since they do not factor in constraints on the number of available data points. Moreover, they do not account for the optimization challenges that arise when changing data distributions.

**Continual learning.** There has been a lot of work on continual learning for new tasks (Rolnick et al., 2019; Parisi et al., 2019). Such works have traditionally focused on reducing catastrophic forgetting (Kirkpatrick et al., 2017) instead of improving target task performance (Gupta et al., 2023; Ibrahim et al., 2024; Kotha et al., 2024; Çağatay Yıldız et al., 2025; Chen et al., 2025a; Springer et al., 2025). There has also been work on methods and evaluation for teaching models new facts (Meng et al., 2023; Yang et al., 2024b; Ghosal et al., 2024; Gekhman et al., 2024; Chang et al., 2024). Our two-stage framework helps build intuition for when pretraining is necessary, as well as shows better ways to teach models new facts. However, answering further questions about capability and knowledge will require using more refined metrics and data distributions.

## 7 DISCUSSION

**Hypotheses for inefficiency of fine-tuning.** In this paper, we thoroughly characterize the failure of fine-tuning. We share two hypotheses for why fine-tuning underperforms replay data. We first identify a training instability that occurs for a few steps of fine-tuning which replay slightly mitigates (experiments and discussion in Appendix B.1). However, even with perfect optimization, we identify a statistical barrier due to a tendency to overfit to small samples. In Appendix B.2, we detail a toy model where failure arises due to a small number of noisy data points, leveraging classical intuition from the double-descent literature. Another reasonable hypothesis is that this is an artifact of our current training scale. To control for this, we ablate model size in Appendix B.3. Interestingly, the benefit of replay seems to persist across model scale, decreasing support for this hypothesis.

**Limitations.** We make a number of necessary simplifications for our controlled setting. For example, we assume two distributions, where as in practice, pre-training is a multi-task learning problem with much higher diversity. The simplicity of our data schedules, though a feature, preclude us from studying more complicated methods such as continuous annealing of the distribution, sample-level orderings, and more advanced fine-tuning methods. Furthermore, we use validation loss, which might not perfectly correlate with downstream metrics. In practice, replay comes at the cost of increased compute, which might be a limiting factor outside of standard fine-tuning settings.

## 7.1 ETHICS STATEMENT

We hope our work can be used to improve the data efficiency of language models, especially for low resource domains that receive relatively less attention. We acknowledge our work may increase the compute used in language model training. We believe most other harms associated with our work are generally applicable to most language modeling research.

## 7.2 REPRODUCIBILITY STATEMENT

We make strong efforts to ensure reproducibility of our results. We will open-source all of our training and evaluation code. In addition, we will provide a WandB report and project page with access to all 1935 runs for different data orders and hyper-parameters.

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

## A  DATA SCHEDULE EQUIVALENCES

As discussed in Section 4.2, data schedules have two degrees of freedom. However, they can be described with many intuitive variables, each a few equations away from each other. We can use the following variables to describe the data schedule:

- **Total training steps** $T$: the total number of training steps.
- **Target step fraction** $\gamma$: the fraction of training steps that are target, after deciding the repetition count.
- **Replay fraction** $\rho$: the fraction of pre-training data that is replayed.
- **Target stage 2 allocation** $\alpha$: the fraction of the total target data that is allocated to Stage 2.
- **Stage 2 duration** $\delta$: the fraction of training that is in Stage 2.
- **Stage 1 target weight** $w_1$: the weight of the target data in Stage 1.
- **Stage 2 target weight** $w_2$: the weight of the target data in Stage 2.

If there are 7 variables, why are there only 2 degrees of freedom? The first two variables are set by problem setting and we do not control them (besides the repetition count, which we treat as fixed for this section). For the rest of this section, without loss of generality, we set $T = 1$. We claim that given the next 2 variables, you can derive the other 3.

If the replay fraction is $\rho$, then the Stage 2 target weight is automatically fixed as $w_2 = 1 - \rho$. If the Stage 2 allocation is $\alpha$, then we know that the number of target steps in Stage 2 is $\alpha\gamma$. This means that the number of total steps in Stage 2 is $\frac{\alpha\gamma}{1-\rho}$, giving the Stage 2 duration $\delta$. Now that we know $\delta$, it suffices to determine the Stage 1 target weight. We know that there are $\gamma(1 - \alpha)$ target steps in Stage 1, as well as $1 - \delta$ total steps. This means that the Stage 1 target weight is $w_1 = \frac{\gamma(1-\alpha)}{1-\delta}$. Therefore, we've recovered all 7 variables from the 2 degrees of freedom. You can confirm that with these choices of $w_1, w_2$ that $w_1(1 - \delta) + w_2\delta = \gamma$.

## B  POTENTIAL FAILURE MODES OF FINE-TUNING

We discuss potential conceptual failure modes of fine-tuning here, using a mix of experiments and toy models for intuition.

### B.1  INSTABILITY OF FINE-TUNING

We notice that at the start of fine-tuning, there is a pretty large loss spike (Figure 11). This is especially true at higher learning rates. However, after some steps of training, the loss goes below where it started. In fact, it is still correct to use higher learning rates even though they make the spike larger because you end up with a lower loss.

One relatively vague hypothesis is that the loss spike is the reason fine-tuning underperforms replay. This can happen in at least two concrete ways

1. When there is replay data, there is less distribution shift between Stage 1 and Stage 2. Therefore, the loss spike is less pronounced, and there is less steps spent recovering from it.
2. There seems to be a minimum number of steps before one recovers from the loss spike. Since replay increases the number of steps in Stage 2, it can perhaps give more time to recover from the loss spike.

We believe further experimentation is necessary to fully understand the nature of this spike, specifically whether it is harmful for model performance and at what data regime it matters the most in.

### B.2  OVERFITTING TO TARGET DATA

It is known that a fixed un-regularized model has a tendency to overfit to small sample counts. Therefore, it's possible that fine-tuning suffers from this problem. To model this, we set up a simple linear regression toy model.

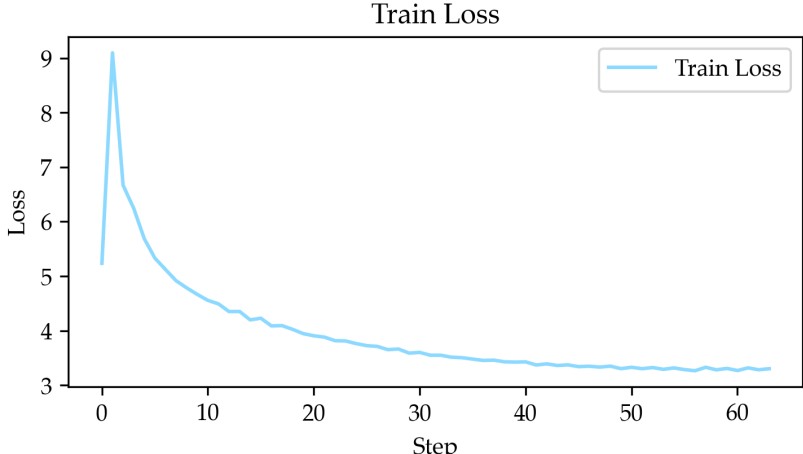

Figure 11: **Train loss spike.** We notice a large loss spike during the first few steps of training across our settings. This represents one barrier to training with a limited number of samples. Replay might help either because (1) there is less distribution shift between Stage 1 and Stage 2 or (2) there is more time to recover from the spike.

We construct a data distribution in $400$ dimensions, governed by a pre-training $\theta_{\text{PT}} \sim \mathcal{N}(0, I)$ and a fine-tuning $\theta_{\text{FT}} \sim \mathcal{N}(\theta_{\text{PT}}, 0.1I)$. We then construct a dataset of $N$ pre-training points and $n$ fine-tuning points. Each point is generated as $(x, \theta^\top x + \epsilon)$ for $x \sim \mathcal{N}(0, I)$ and $\epsilon \sim \mathcal{N}(0, 1)$.

Our training algorithm first "pre-trains" by performing OLS on the pre-training points. For sufficiently large $N >> d$, this will easily fit the pre-training vector $\theta_{\text{PT}}$. However, our goal is to learn the fine-tuning vector $\theta_{\text{FT}}$. We do this by performing OLS on the residuals of the pre-training fit and the true labels for the fine-tuning points. The fine-tuning now benefits from the pre-training fit bringing the parameters closer to the true fine-tuning vector. When we are in the over-parameterized regime, we use min-norm least squares regression, as double-descent literature has shown this is known to generalize better and is reflective of deep learning inductive bias (Belkin et al., 2019; Hastie et al., 2020). We then measure error as mean squared error between the learned parameter and true fine-tuning parameter. We visualize these results in Figure 12, purple line. We see that for $n < d$, the model overfits to the noise in the fine-tuning data, resulting in higher error than random guessing. We plot the gray line to track the best possible error achievable by the model for a given $n$ by using any sample count up to $n$.

We now introduce replay: mix in some pre-training data for the fine-tuning OLS. The replay significantly reduces the overfitting as we see the in the other lines.

In this setting, it is known that the Bayes-optimal estimator involves appropriately tuning the ridge regularization parameter. We visualize differing values of the ridge parameter in Figure 13, orange line. We see that the optimal ridge parameter is non-zero. Moreover, the best ridge parameter achieves much better loss than tuned replay count. If this intuition holds, real language model training should benefit from finding the correct notion of regularization. One might expect the natural analogue of ridge regression for language models is weight decay. As discussed in E.1.3, weight decay does not significantly improve the loss and under-performs optimal replay. This tells us we need to rethink how we regularize fine-tuning to extract the full value of target data points.

### B.3 MODEL SIZE

One concern is that the necessity of replay is due to the model size. To see whether any explanation related to model size holds ground, we see whether changing model size changes the necessity of replay. To measure this, we take our joint training setting with a fixed learning rate schedule and $\alpha = 1.0$ (all target data in Stage 2) and vary the model size. When we increase model size, we decrease the learning rate inversely proportionally to the width of the hidden dimension, following

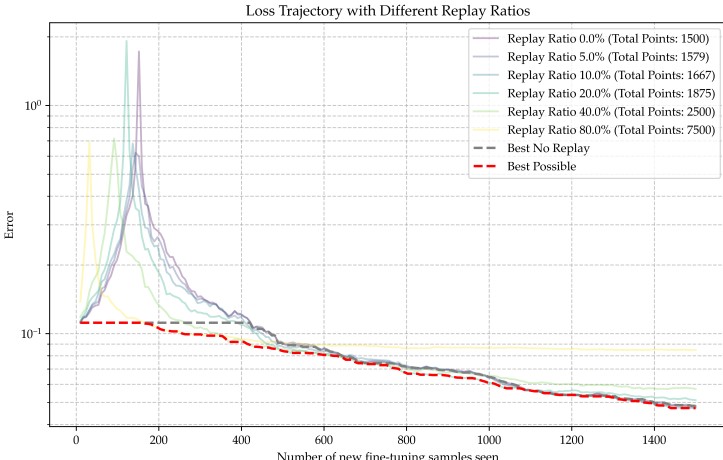

Figure 12: **Linear regression with replay.** We plot the loss of a linear regression model as a function of the number of fine-tuning points $n$ for different values of the replay fraction $\rho$. We see that for $n < d$, the model overfits to the noise in the fine-tuning data, resulting in higher error than random guessing. We plot the gray line to track the best possible error achievable by the model for a given $n$ by using any sample count up to $n$. Replay significantly reduces the overfitting, resulting in better MSE than random guessing. We track the best possible error as a function of $n$ for the best $\rho$ as the red line. For a regime of some but not too many target points, replay improves over standard fine-tuning.

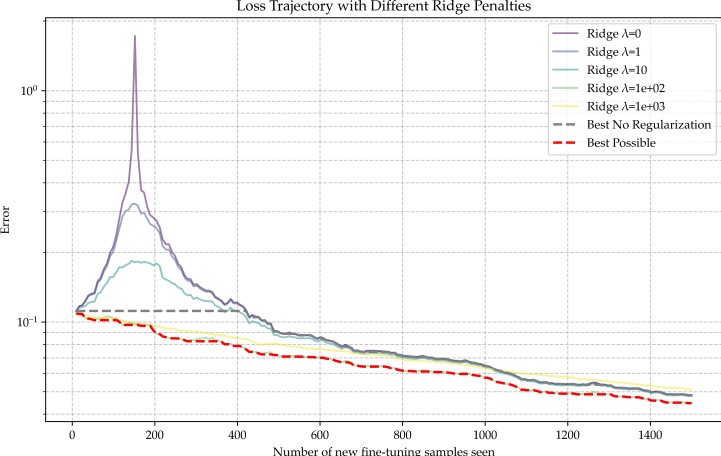

Figure 13: **Linear regression with ridge regularization.** We follow the same setup as in Figure 12, but instead of tuning replay fraction, we tune the ridge regularization parameter $\lambda$. We see that the optimal ridge parameter is non-zero and that the best ridge parameter achieves much better loss than tuned replay count.

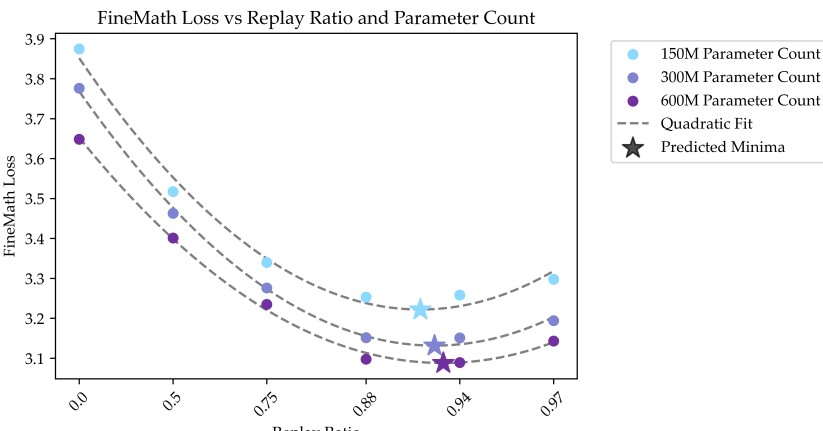

Figure 14: **Model scaling.** We take our standard 4B token training setup and scale the parameter count to 4x parameter count. For a given size, we sweep for the optimal replay fraction $\rho$ while reserving all the target data for Stage 2 ($\alpha = 1.0$). We find that larger models still require replay data to get lower loss.

folklore scaling practices (Everett et al., 2024). We visualize the results in Figure 14. We see that across all sizes, the model benefits from replay. Furthermore, the benefit of replay is relatively consistent across model sizes.

One interesting implication of this finding is that one can determine the optimal data schedule for a large model by tuning on a small model. This resembles $\mu$P style arguments (Yang et al., 2022; 2024a) for setting layer-wise learning rates at small model counts.

## C   NOTE ON TUNING APPENDICES

This project has spanned a lot of experiments. The order these were conducted in does not reflect the order in which they are presented. At a high level, most of the mid-training/pre-training experiments were done first, giving intuition for what the results would look like for supervised fine-tuning. This means the experiments are more comprehensive for mid-training as it was when there was a worse understanding of the relationship between problem parameters. As the project developed, we were able to reduce search spaces by good priors on what hyperparameters worked (though we always certified they were correct as shared in the plots).

We believe it is more instructive to read the guide for how to set mid-training hyperparameters. We believe the literature lacks rigorous experiments (mostly deciding random hypers on the fly). However, this stage can give large data efficiency gains if done correctly.

## D   GENERAL TRAINING SETTINGS

We train a 150 million parameter Llama-style language model with context length 4096 for 4B tokens. This is close to Chinchilla optimal scaling (Hoffmann et al., 2022) which prescribes 20x tokens per parameter. We train with batch size 1024 for 1024 steps with weight decay 0.1. We use the Adam optimizer with default parameters. When we tune learning rate, we search for the nearest power of 3 assuming the final loss is convex in the learning rate. For the other models, we scale the learning rate inversely with the model width following the reccomendations of (Everett et al., 2024) (see Table 1).

For our generic data, we use C4 (Raffel et al., 2023) since it is filtered but relatively uncurated. For example, C4 filters out all code data by removing documents with curly braces. For our target data, we have domains representing math (FineMath (Allal et al., 2025)), coding (StarCoder (Li et al., 2023)), and instruction following (Flan (Longpre et al., 2023)). Our validation datasets are always the same distribution as our training data.

## D.1 Model configurations

| Parameter | 150M | 300M | 600M | 1.9B |
|---|---|---|---|---|
| Context Length | 4096 | 4096 | 4096 | 4096 |
| Hidden Dimension | 512 | 768 | 1024 | 2048 |
| Intermediate Dimension | 1792 | 2688 | 3584 | 7168 |
| Attention Heads | 8 | 12 | 16 | 16 |
| KV Heads | 8 | 12 | 8 | 8 |
| Layers | 6 | 12 | 24 | 24 |
| Learning rate | 3e-3 | 2e-3 | 1.5e-3 | 7.5e-4 |

Table 1: Model architecture configurations for different model sizes. All models use the Llama architecture with a standardized context length of 4096 tokens. We default to the 150M model if not specified.

## D.2 Magic number justifications

Selecting a training regime requires setting some arbitrary numbers. We give justification for some here.

- Target data fraction: Pre-training token counts are on the order of 10 trillion while domains are on the order of 10 billion tokens, motivating our choice of $\approx 0.1\%$. We actually use a target data fraction of $\frac{1}{1024}$; this better interacts with our block-deterministic data scheduler which draws/shuffles 2048 sequences at a time.

- Replay fractions and target data allocations: In early experiments we quickly realized that the dependence on these parameters scaled nicely when the replay fractions are spaced out by $\log(1-x)$. Therefore, we equally spaced out values. In plots, we round all values to two decimals. In actuality, our replay fractions were $0.25, 0.5, 0.75, 0.875$ and our target data allocations were $1.0, 0.5, 0.25, 0.125$. The power of 2 scaling similarly interacts nicely with our block-deterministic data scheduler.

- Model size: 150M reflects a model scale that is large enough to be represantative and scale nicely. It also enabled quicker iteration than larger scale models. During the course of this project, we sanity checked our results held at larger scales, increasing our confidence in using smaller scale models.

# E Mid-training experiments

## E.1 Fine-tuning baseline

### E.1.1 Repetitions

We try varying the number of repetitions of the target data during mid-training. For this tuning, since we do not know the learning rate and schedule yet, we tune across the two most promising learning rates of 1e-3 and 3e-3 with no learning rate cooldown (as this is closer to our final learning rate schedule than full decay). We visualize the best of both in Figure 15. We find that we can tolerate up to 32 repetitions of the target data before overfitting across all domains.

### E.1.2 Learning rate cooldown

We vary the cooldown duration of a standard WSD learning rate schedule with 10 step warmup while fixing all 32 repetitions of the target data to appear at the end of training. We also vary the learning rate to be 1e-3, 3e-3, and 1e-2. For all training runs, 3e-3 did best and 1e-2 always diverged, so we can safely only look at WSD with learning rate 3e-3. We visualize the final result in Figure 5. We find it critical to have a cooldown period as opposed to the more conventional long cooldown period (cooldown duration 0.99).

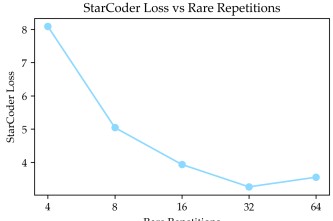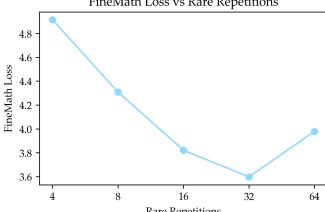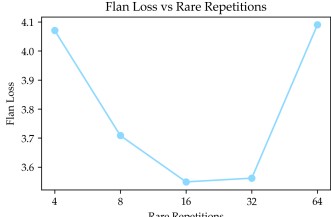

Figure 15: **Mid-training tuning repetitions.** We show that across our mid-training domains, we can tolerate maximum 32 repetitions of the original data before overfitting to the target data. Note that the x-axis is more comprehensive compared to 18.

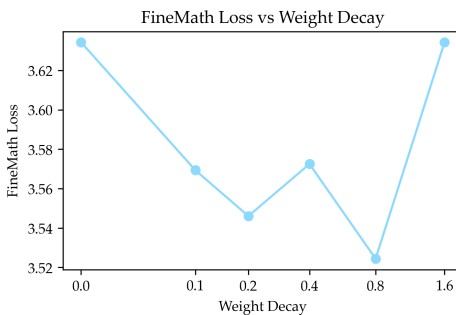

Figure 16: **Tuning weight decay.** We tune the weight decay for the fine-tuning baseline fixed to our above repetition count and learning rate cooldown. The effect is small and noisy, so we default to 0.1, at the upper range of weight decays used in the literature.

### E.1.3 TUNING WEIGHT DECAY

We try tuning the weight decay for the fine-tuning baseline fixed to our above repetition count and learning rate cooldown. We visualize the results in Figure 16. We find that weight decay does give minimal but noisy effect on loss. Due to this variance, we decide to stay closer to the range of weight decays used in the literature which are usually around 0.05 (for example, Table 10 and 11 in the appendix of Li et al. (2025)) However, since there does seem to be a benefit of slightly higher weight decay, we use 0.1 for all of our mid-training experiments. We also carry this choice to our pre-training experiments.

## F CHARACTERIZING FORGETTING

In addition to characterizing loss for the target domain, we also measure the loss on the generic domain to quantify how different data schedules result in different amounts of forgetting. In Figure 17, we show how the loss changes across data schedules, similar to Figure 7. We find that both introducing replay data and target data early significantly mitigate forgetting.

## G POST-TRAINING EXPERIMENTS

### G.1 FINE-TUNING BASELINE

#### G.1.1 REPETITIONS

We try varying the number of repetitions of the target data during fine-tuning. We visualize the loss for different repetition counts in 18. We find that we can tolerate up to 64 repetitions of the target data before overfitting across all domains.

#### G.1.2 LEARNING RATE

For the model pre-trained on C4, we tried different learning rates for 1000 training steps and picked the model with the best C4 loss. This ended up being 1e-3 (which acheived 4.12 loss, relative to 3e-3 which achieved 4.22 and 3e-4 which achieved 4.66). This learning rate was used across all pre-trained

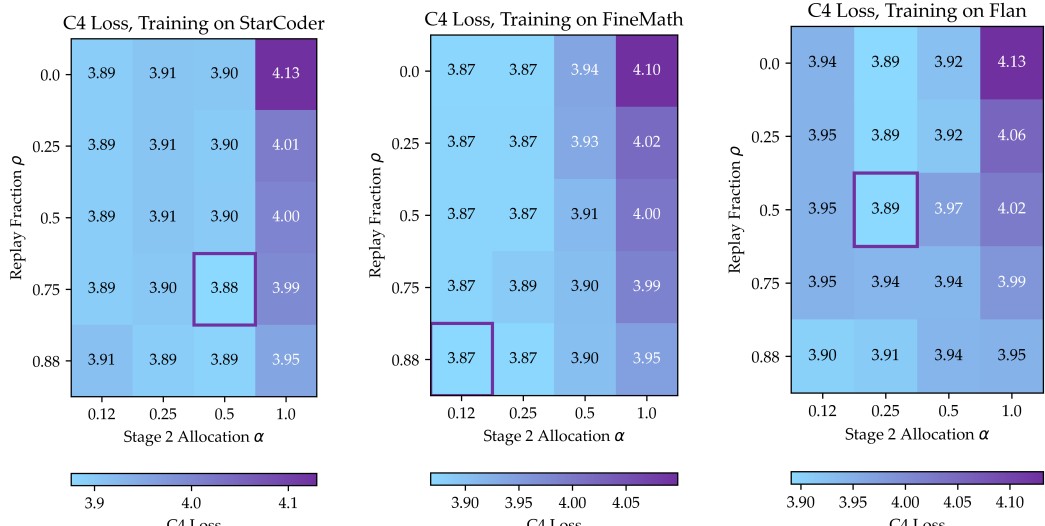

Figure 17: **Full data schedule sweep for forgetting.** We take the same data schedules in 7 and instead plot loss on the generic domain (C4) instead of loss on the target domain, quantifying the amount of forgetting. We find that replay and introducing target data early both significantly reduce forgetting.

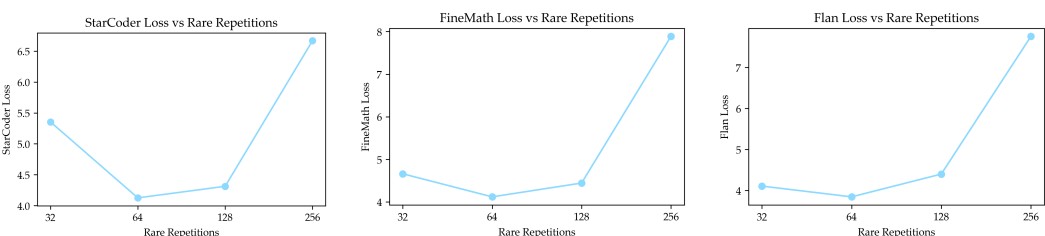

Figure 18: **Fine-tuning repetitions.** We show that across our fine-tuning domains, we can tolerate 64 repetitions of the original data before overfitting to the target data.

models, which ranged from 512 to 992 steps. Note that the optimal learning rate is different for this setting compared to the mid-training experiments because we're using a different learning rate.

## H  DATA EFFICIENCY

It is important to compare how well two training algorithms leverage the same amount of fixed samples. Though we can compare the direct loss, this gives a misleading impression of how what the actual improvement is. To bring this into a human-friendly metric, we introduce the data efficiency multiplier. We use the following procedure:

1. **Fix a reference training algorithm.** To enable comparison across a broad suite of possible training strategies without having to refit scaling laws, we first fix a reference training algorithm $S_{ref}$ that characterizes one natural usage of the training data. We detail our choice of reference algorithm below.

2. **Build a power law.** We now build a reference scaling law to characterize the loss of the reference algorithm as a function of the number of target tokens it gets to see. Since the model size is fixed, we fit a power law from number of data points seen to loss (more details below).

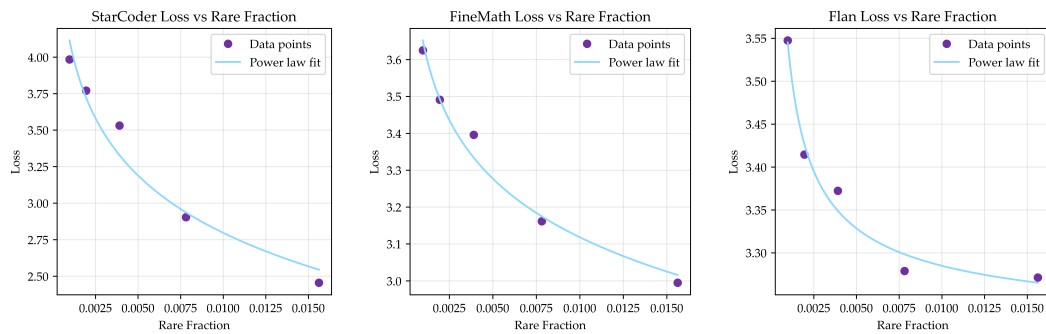

Figure 19: **Power law fit.** We fit a power law to the loss of the reference algorithm (uniform mixing) as it receives more target data. We note that we use these laws in the interpolation regime since we train models with hypothetically $16\times$ more data than we actually train with.

3. **Effective data points.** For any given strategy S, we can find it's loss. With this loss, we can see how many data points $D(S)$ it would take for the reference algorithm to match the loss of S. If this number is high, then S is very data efficient.

4. **Normalize for reference.** However, our current metric strongly depends on the choice of reference algorithm. To make this metric useful regardless of the reference algorithm, we always report a *data efficiency improvement*. Specifically, we only use this metric to compare the data efficiency of two strategies $S_1$ and $S_2$. We then report the data efficiency improvement of $S_2$ over $S_1$ as $\frac{D(S_2)}{D(S_1)}$. A large improvement means that $S_2$ is much more data efficient than $S_1$.

We now go into details about how we fit the power law.

## H.1    POWER LAW FORMULATION

We fix the model size and total number of tokens. We then fit a power law from number of target data points given to loss. Our power law has the form $L(D) = aD^b + c$ for loss $L$, number of target data points $D$, and free variables $a, b, c$. We use `scipy.optimize.curve_fit` to fit the scaling law.

## H.2    TRAINING RUNS

We fix learning rate schedule to be cosine and tune learning rate to be 0.003. When tuning epoch count, we found that the model could not tolerate more than 32 epochs of target data at larger target data fractions. Therefore, we fix this epoch count. We also mix data uniformly throughout training instead of using a dedicated data schedule. We train for a total of 4B tokens and vary the number of target tokens to be 4M, 8M, 16M, 32M, 64M tokens. Since we train for 32 repetitions, our largest target data run trains on target tokens for $50\%$ of training.

There is some extra noise in these fits compared to our other experiments since we can not control for data order when we change the target fraction. However, we note that the best training runs for the reference algorithm with extra data outperform the best data orders for the low data fraction we fix throughout the paper. Fortunately, this keeps us within the interpolation regime of the scaling law and it doesn't matter whether this scaling law extrapolates past the data fractions we train with. We plot the fit in Figure 19.

# I    WSD TUTORIAL

It turns out that the correct learning rate schedule is critical for improving target data efficiency. Though annealing-based learning rate schedules give the largest benefit for data ordering, we have relatively little intuitive/empirical understanding of how to properly use them. We will provide a quick introduction to how they work in the un-ordered setting, and then show how to use them in the ordered setting.

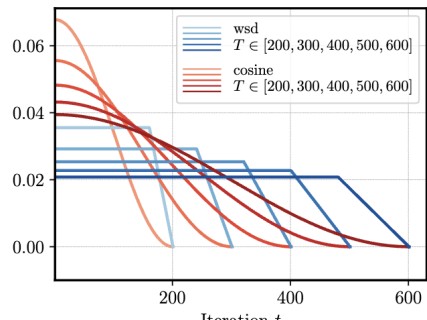

Figure 20: **Learning rate schedule.** This figure shows the shape of a WSD learning rate schedule in contrast to a cosine learning rate schedule (both without warmup). Figure is taken from Schaipp et al. (2025), Figure 2 left.

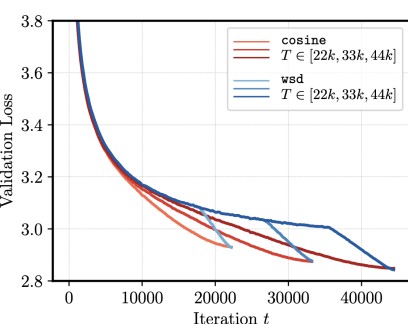

Figure 21: **Loss curves.** This figure shows the loss curves for a WSD learning rate schedule and a cosine learning rate schedule. Note that the loss of a WSD schedule initially makes slower progress than a cosine schedule and then makes up for it with much faster loss improvement at the end. Figure is taken from Schaipp et al. (2025), Figure 1 left.

## I.1 WHAT IS WSD?

Warmup-Stable-Decay (WSD) is a learning rate schedule with three phases:

1. Warmup: The learning rate is increased linearly from 0 to a peak value.

2. Stable: The learning rate is held constant at the peak value.

3. Decay: The learning rate is decayed linearly from the peak value to 0.

We visually depict this learning rate schedule (without warmup) in Figure 20, left.

## I.2 STANDARD TRAINING (RANDOM ORDER)

Though warmup is important, the exact duration of the warmup is not critical. In contrast, the decay period is critical to the final loss. We visualize the loss curves for a WSD learning rate schedule and a cosine learning rate schedule in Figure 21, right. Notably, as soon as the learning rate starts decaying, the loss improvement is much faster. This is contrast to cosine learning rate schedules where the loss improvement actually slows down at the end of training (with a characteristic curl up). These different rates of decrease have historically been really important details: for example, fitting the scaling laws to intermediate checkpoints gives incorrect scaling laws since the models have been annealed for different durations (Hoffmann et al., 2022).

Why does this happen? One nice intuitive picture is given by the river valley landscape explanation in Wen et al. (2024). They posit that the loss landscape looks like a single river flowing down in the middle of a valley (Figure 22). In this picture, you would like to both get to the bottom of the valley and go far down the river. A standard learning rate schedule slowly descends the valley while also making progress along the river direction. The paper's central claim is that WSD instead stays at the top of the valley but continues to make progress along the river direction. Then, when one anneals the learning rate, it starts descending donw the valley, revealing the true progress made by the model not captured by the loss. It is noted in the Edge of Stability literature (Cohen et al., 2022; 2024) that staying at high learning rate is better for performance even though there is large oscillation in the loss.

Though this picture is helpful visually, there is an even simpler theoretical picture. The works Defazio et al. (2024); Schaipp et al. (2025) show that the simple theoretical model of non-smooth convex optimization predicts the shape of the loss curve. Specifically, the upper-bound on the loss from

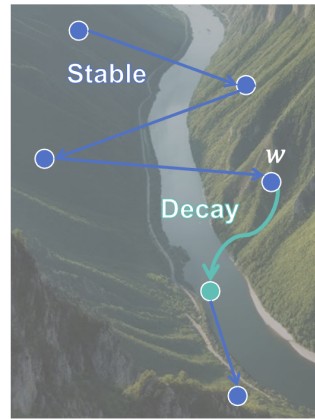

Figure 22: **River valley landscape.** This shows the intuitive picture of the river valley landscape. WSD makes progress along the river direction while making all the hill progress at the very end. Figure is taken from Wen et al. (2024), Figure 2 left.

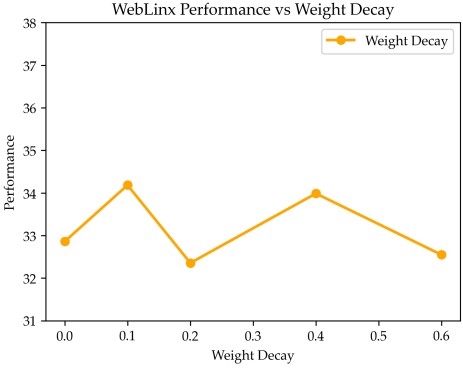

Figure 23: **Weblinx weight decay ablation.** We fine-tune Llama 3.1-8B Instruct on Weblinx demonstrations. We find that without replay, tuning weight decay gives little gains, improving by less than $2\%$ over the baseline.

standard online convex optimization arguments applied to the last iterate of training matches the "shape" of the WSD loss curve.

### I.3 ORDERED TRAINING

Why does it matter that WSD decreases loss faster at the end of training? Intuitively, if the loss is decreasing faster, placing high quality data at the end of training is more important. We algorithmically leverage this intuition by placing the target data at the end of training with WSD. In some earlier experiments, we found that when using a cosine learning rate schedule, it actually hurt to keep target data at the end of training relative to placing it uniformly throughout training.

## J WEB AGENTS

We train with the same hyperparameters as the original Weblinx paper on a subset of the demonstrations from the original paper. We use the same evaluation protocol and metrics as the original paper by combining the validation and in-distribution test set. We defer to the original paper for more details on the data and evaluation. When we specify replay fraction, we are doing the replay on a document level instead of a token level. This doesn't have large implications since all peak at an intermediate value and fine-tuning isn't computationally intensive.

We provide an additional ablation on tuning weight decay while fine-tuning on web agents data. We find that without replay, this gives little gains, improving by less than $2\%$ over the baseline. We provide the results in Figure 23.

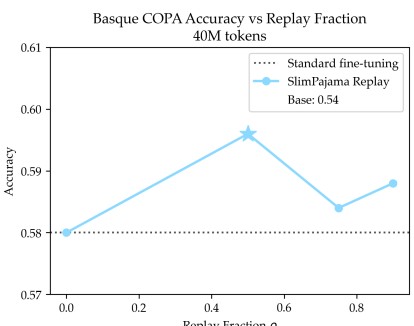 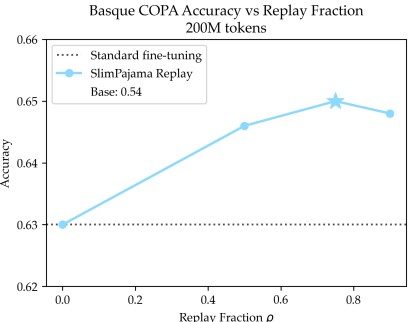

Figure 24: **Basque training with different token counts.** We try Basque training with 40M and 200M tokens to confirm that the gain in accuracy is real.

## K  BASQUE

We tune the learning rate to be 1e-5 for fine-tuning on Basque. We find the gain in accuracy to be real across different token counts, displayed for 40M tokens and 200M tokens in Figure 24.

When tracking Basque loss, we find that there is a spike in loss at the start of training only if the learning rate is sufficiently high. In practice, it is worth using training with this higher learning rate for best Basque loss/accuracy. We find that the loss improvement of replay decreases as we increase the total token count. However, there continues to be an accuracy gain as you increase the token count, showing that replay may be even more important for evaluation metrics.

## L  DETAILED RELATED WORK

### L.1  REPEATING DATA

Prior work on data-constrained scaling laws (Muennighoff et al., 2023; Goyal et al., 2024) predict that as you continue to repeat data, loss improves at a diminishing rate. However, the specific decay formulation predicts that it will asymptote at a particular value.

Specifically, the simplest decay formulation presented in both works estimates that when you see $n$ data points for the second time, it is like effectively training on $n\delta$ data points for decay factor $\delta$. For the $k$-th repetition, it is like training on $n\delta^{k-1}$ data points. In the infinite data limit, these scaling laws predict that the loss will asymptote at a particular value, specifically at the loss of seeing $\frac{n}{1-\delta}$ fresh data points once.

In our experiments, we found this not to be the case. If we repeated a target domain too many times, the loss would start going up. This is true whether it is fine-tuning or it is generic pre-training data. This means we think more carefully about how to leverage target data. This observation is corroborated in (Kim et al., 2025).

### L.2  NECESSITY OF PRETRAINING

Many prior works argue that it is necessary to incorporate target skills during pretraining. For example, (Allen-Zhu and Li, 2024; Jiang et al., 2024b) argue that instruction-tuning data needs to be seen during pretraining. Moreover, many practitioners pretrain language models from scratch with the belief that it is necessary to see this data during pretraining. Our work shows that this might not be the case: for some tasks, data might not need to be seen during pretraining, as long as one follows optimal training procedures for adaptation.

### L.3  ROBUST FINE-TUNING

There is a rich literature on how to robustly fine-tune language models to maximize in-distribution and out-of-distribution accuracy (Phang et al., 2019; Zhang et al., 2021; Kumar et al., 2022). Weight

averaging has been one such technique to improve post-training performance (Wortsman et al., 2022; Ilharco et al., 2023; Dang et al., 2025). Replay can be seen as qualitatively similar to weight averaging where the averaging takes places in data distribution space instead of parameter space. In contrast to prior work, we characterize the interaction between pre-training and fine-tuning, showing that the optimal way to fine-tune depends on how much exposure the pre-trained model has to the target task. Moreover, since the focus was primarily out-of-distribution performance, they under-focused on the opportunity to improve in-distribution performance.

## L.4 CURRICULUM LEARNING

Curriculum learning is concerned with proposing a sequence of training distributions from easy to hard (Bengio et al., 2009). Theoretically, this can accelerate convergence by introducing tractable intermediate tasks (Abbe et al., 2023; Panigrahi et al., 2024). Recent works have tried to design curricula using reference models (Mindermann et al., 2022; Fan and Jaggi, 2023; Lin et al., 2025) or structure over the data distribution (Chen et al., 2023). However, there is limited evidence that changing data order improves the final performance of models on tasks in iid settings (Wu et al., 2021). In contrast, our work focuses on the *relevance* of the data with respect to the target task, where it is well known that changing data order improves performance (e.g. fine-tuning).

