# OpenReview forum: "Scheduling data improves fine-tuning data efficiency"
_ICLR.cc/2026/Conference — ICLR 2026 Conference Withdrawn Submission_

### Official Review · Reviewer_p7u4 · 2025-10-24

**Soundness:** 2
**Presentation:** 2
**Contribution:** 2
**Rating:** 4
**Confidence:** 4

**Summary:**

The paper proposes a new method to improve the data efficiency for mid-training scenarios. They show a simple yet effective method to improve performance on target domain and use mid-training with a single learning rate schedule as a baseline. After that, they search a best configuration of the ratio of target data during pre-training phase to achieve an up-to 15.86x data efficiency improvement.

**Strengths:**

1. The research question is pivotal for low-resource training.
2. Experiment results (up to 15.86x fewer data) shows the superiority of the proposed method.

**Weaknesses:**

1. While the paper presents compelling empirical results, several limitations undermine its overall contribution. First, the novelty is limited: the proposed “best” scheduling strategy is essentially a combination of two previously established ideas—(i) replaying generic data during fine-tuning (a standard technique in continual learning) and (ii) using a unified Warmup-Stable-Decay (WSD) schedule with early target data injection (as in recent mid-training works). The paper does not introduce a new algorithmic component or theoretical framework, but rather demonstrates that these existing interventions are complementary—a useful but incremental finding.
2. Second, the method lacks robustness and practicality: it introduces multiple sensitive hyperparameters (e.g., $\rho,\alpha,\gamma$), and performance varies significantly with their choice (see Figure 7). Crucially, the authors provide no principled or automated way to select these parameters, which severely limits applicability in real low-resource settings where validation data for tuning may be scarce. Figure 7 shows that the selected hyperparameters are robust, while the selected hyperparameters in Flan is not as robust as that in FineMath. How to select such hyperparameters should be further investigated.
3. Third, the evaluation is narrowly scoped to Llama-style architectures, raising concerns about generalizability. Different model families (e.g., Gemma, Phi, or MoE-based models) may respond differently to data scheduling due to variations in normalization, activation functions, or attention mechanisms, yet no experiments on alternative backbones are reported.
4. Finally, the mechanism behind the key finding—that replay improves in-distribution target performance—remains poorly understood. The toy linear regression model suggests overfitting mitigation via implicit regularization, but this contradicts the main experiments where explicit weight decay fails to replicate replay’s benefits (Appendix E.1.3). Without deeper analysis (e.g., gradient dynamics, representation alignment, or loss landscape geometry), it is unclear whether the gain stems from optimization stability, statistical regularization, or another factor, limiting the work’s theoretical insight.

**Questions:**

see weakness

---

### Official Review · Reviewer_5WEY · 2025-10-30

**Soundness:** 3
**Presentation:** 4
**Contribution:** 2
**Rating:** 4
**Confidence:** 4

**Summary:**

This work presents a detailed investigation into how much we can improve performance on the target domain  with a limited amount of data via adding generic data to the end of training or target data to the start. Specifically, it constructs a controlled pre-training environment and adjusts the proportion of replaying generic data in the fine-tuning stage and target data in the pre-training stage. This proves that both strategies are helpful to improve the data efficiency. Furthermore, experiments on agentic web navigation and Basque question-answering demonstrate that replaying data is rather useful in real-world applications.

**Strengths:**

1. It systematically explores the potential reasons of fine-tunineg failure.
2. It compares different data and learning-rate-based training schedules, concluding that standard fine-tuning < replaying generic data < mid-training baseline < two-stage data schedules.

**Weaknesses:**

1. This paper proposed 3 strategies (replay, mid-training, allocating part of target data to pre-training stage) to improve fine-tuning performance. However, it seems that only the first one is applicable given a pre-trained LM, which has been well explored in related works in line 452. For the other two strategies, it is too expensive to pre-train from scratch just for a target model with limited target data.
2. This paper only describes the phenomenons but cannot provide applicable guidance to practice. For example, how can we choose \alpha and \rho for a specific target domain given the amount of available target data? Besides, what kind of generic data is preferred for a specific domain?
3. Not enough analysis of the inherent reasons. Why can those strategies (replaying etc.) boost target domain performance? Where is the performance boundary?
4. I’m not sure the measurement of data efficiency is reasonable because it cannot be directly observed from the figures and the improvements seem to be much more significant than loss. For instance, in Figure 7, you describe that data efficiency improvement for Flan by 2.06x and 4.80x with generic replay and two data schedules respectively, but the loss only degrades from 3.44 to 3.37 and 3.29 respectively.

**Questions:**

1. What is the specific definition of “domain with a limited data”? There are abundant datasets for math, code, and instruction-following. If the amount of data is too small, then it is hard to allocate part of them to pre-training stage.
2. What is “Rare Fraction” in Figure 2?
3. Does the “pre-annealed pre-training checkpoint” in line 257 mean the checkpoint that stops before learning rate decays?
4. Do the generic datasets (e.g., OpenHermes and UltraChat) appear in the pre-training dataset? If not, it seems that you can not call this behavior as “replay”.
5. When replay ratio \rho >0, the fine-tuning steps always grows with 1/(1-\rho). Is it reasonable to calculate the data efficiency as described in Section 2.2 with more training steps/data (even replaying)?

typos:

1. reversed descriptions of left and tight in lines 178-179.
2. What does “Based: 0.54” mean in Figure 10?

---

### Official Review · Reviewer_3Pgi · 2025-10-31

**Soundness:** 2
**Presentation:** 2
**Contribution:** 2
**Rating:** 2
**Confidence:** 3

**Summary:**

This paper investigates how the scheduling of data—that is, the order and mixture of generic and target-domain data—can dramatically improve the data efficiency of fine-tuning language models. The authors challenge the standard paradigm of pre-training on a massive generic dataset and then fine-tuning exclusively on a limited target dataset. They systematically show that this standard approach is highly inefficient and propose several interventions that yield significant performance gains with the same amount of target data.

**Strengths:**

The paper's primary strength in originality lies in its reframing of the fine-tuning problem. It shifts the focus from what data to use, to when and how that data should be scheduled.

 The authors conduct their investigation in a highly controlled environment, starting with a smaller model to systematically and affordably sweep the hyperparameter space. This scientific approach allows them to isolate the effects of their interventions with high confidence.

The paper builds its argument layer by layer, starting with the simple intervention of replay, moving to the unified mid-training baseline, and culminating in the full two-stage schedule. This structure makes the final, complex result feel like a natural and well-motivated conclusion.

**Weaknesses:**

This is a well-executed and clearly written paper, but there are several limitations regarding the generalizability, practical applicability, and novelty of the findings that temper its overall impact.

1.Limited Generalizability of the Headline Claim: The most striking result—a 15.86x improvement in data efficiency—is demonstrated in a highly controlled, small-scale environment (a 150M parameter model). While the authors commendably validate that replay is helpful at a larger scale, the magnitude of the benefit is far more modest (2-4.5% accuracy gain). This creates a disconnect between the paper's headline claim and the results shown in more realistic settings.
（1）Reliance on a Proxy Metric: The 15.86x figure is based on inverting a scaling law for validation loss, not a direct, end-task metric like accuracy on math problems. While loss and accuracy are correlated, they are not the same. It is not demonstrated that a 16x reduction in data would yield equivalent task-solving capability.
（2）Simplistic Data Distributions: The experiments rely on a clean separation between generic data (C4) and distinct target domains (math, code, instructions). In many real-world scenarios, pre-training data already contains traces of the target domain, and the distributions are far more overlapping and noisy. It is unclear how these scheduling benefits would change in such "messier" data environments.

2.Impracticality of the Optimal Strategy: The paper's most effective data schedule requires modifying the pre-training stage to introduce target data early. This is a significant practical barrier for the vast majority of AI practitioners, who do not train models from scratch but rather fine-tune existing, pre-trained foundation models. The most actionable advice for this large audience is simply to mix in generic data during fine-tuning (replay). While useful, this is a less novel contribution and yields substantially smaller gains, thus limiting the paper's practical impact for the broader community.
3.Incremental Novelty of the Core Mechanisms: The individual components of the proposed solution, while combined effectively, are not new.
（1）Experience Replay: This is a cornerstone technique from the field of continual learning, where it has been used for years to mitigate catastrophic forgetting. The paper's novelty lies in observing its benefit for the new task, which is an interesting but not paradigm-shifting finding.
（2）Curriculum Learning: The idea of ordering data to improve training efficiency is also a well-established concept. The proposed two-stage schedule can be viewed as a simple, discrete form of a curriculum.The primary contribution is therefore not a new algorithm, but rather a strong empirical demonstration and systematic study of how to best combine these known techniques in the context of fine-tuning, which lowers its fundamental scientific novelty.

4.Oversimplification of the Data Schedule: The paper parameterizes the entire space of data schedules into a two-stage model defined by two variables (α and ρ). While this is an excellent simplification for a systematic search, it is unlikely to be the true optimal schedule. A more continuous, gradual shift in the data distribution from generic to target might yield even better results and represent a more realistic training process. The paper stops short of exploring these more complex curricula.

**Questions:**

1.Regarding the main claim of 15.86x data efficiency on FineMath: Could you report the corresponding improvement on an end-task metric, such as accuracy on a held-out set of math problems? It would be crucial to understand if the significant gains in validation loss translate proportionally to gains in actual problem-solving ability.

2.Your model scaling experiments in Figure 14 suggest that the benefit of replay is consistent across model sizes up to 600M. Do you have any intuition or evidence on whether the optimal schedule (i.e., the best values for  α  and ρ) remains stable or shifts as you scale to much larger models, such as 70B or beyond?

3.Given that modifying pre-training is impractical for most users, what would you recommend as the best "post-training only" strategy? The Llama 3 experiments show that simple replay is effective, but could this be improved? For instance, did you experiment with using a different, more targeted dataset for replay (e.g., general instruction data instead of web text) when fine-tuning for a specific instruction-based task?

4.The two-stage data schedule is a discrete simplification. Have you considered a more continuous schedule, for example, by slowly annealing the proportion of target data from 0% to 100% over the course of training? How do you think such a strategy would compare to the two-stage approach?

5.The paper makes a compelling case for using a single, unified WSD learning rate schedule. How sensitive are the results to this specific choice? If you were to use a more traditional cosine schedule throughout the unified training process (without an optimizer reset), how much of the data efficiency gain would be lost?

---

### Official Review · Reviewer_rAPX · 2025-11-01

**Soundness:** 3
**Presentation:** 2
**Contribution:** 2
**Rating:** 4
**Confidence:** 3

**Summary:**

This paper challenges the standard "pre-train then fine-tune" paradigm, demonstrating it is highly inefficient for adapting models to target domains with limited data. The authors show that replaying generic pre-training data during the fine-tuning phase surprisingly boosts performance on the target task, acting as an effective regularizer. They further find that merging pre-training and fine-tuning into a single, continuous run with a unified optimizer state and a Warmup-Stable-Decay (WSD) learning rate schedule is vastly more efficient. By also moving some target data earlier into the training process, their best-proposed data schedule achieves an estimated $15.86 \times$ improvement in data efficiency over the standard baseline in a controlled setting. The practical value of replaying generic data is confirmed at scale, improving Llama 3 8B's performance on web navigation by $4.5\%$ and low-resource Basque question-answering by $2\%$.

**Strengths:**

1. The paper conducts a rigorous and extensive empirical study in a controlled environment to systematically compare the data efficiency of various training schedules.
2. The key finding that replaying generic data improves target task performance is validated on large-scale models like Llama 3 8B, demonstrating practical, measurable gains on real-world tasks.
3. The work provides a clear and actionable recommendation for practitioners by establishing a much stronger "mid-training" baseline and quantifying the significant inefficiency of the standard fine-tuning paradigm.

**Weaknesses:**

1. Compute Cost of Replay: The compute overhead of replay (increased training steps) is acknowledged but not quantitatively compared to the data efficiency gains.
2. Limited theoretical foundation: The explanations for why replay helps (distribution shift smoothing, overfitting mitigation) are intuitive but hand-wavy. The linear regression toy model is too simplified to provide deep insight. Why does replay specifically help versus other forms of regularization?
3. The paper's central idea of mixing two data sources is an incremental engineering exploration rather than a significant conceptual innovation.

**Questions:**

1. Can you provide analysis showing compute costs vs. performance gains? When is replay worth the additional compute?
2. How do your schedules compare to other gradual domain transition approaches in the literature?
3. Could you ablate how much of the improvement comes from WSD versus simply avoiding the optimizer reset?

---

### Author Response · Authors · 2025-11-21

Thank you to all of the reviewers for their helpful feedback! We will work to improve this paper for a future submission and will withdraw the paper from this cycle.

---

### Note · Authors · 2025-11-21

I have read and agree with the venue's withdrawal policy on behalf of myself and my co-authors.